# The diagnostic accuracy of digital PCR, ARMS and NGS for detecting KRAS mutation in cell-free DNA of patients with colorectal cancer: A systematic review and meta-analysis

Peng Ye[1]*, Peiling Cai[1], Jing Xie[2], Yuanyuan Wei[3]*

1 Department of Anatomy and Histology, School of Preclinical Medicine, Chengdu University, Chengdu, Sichuan Province, People's Republic of China, 2 Department of Pathology and Clinical Laboratory, Sichuan Provincial Fourth People's Hospital, Chengdu, Sichuan Province, People's Republic of China, 3 Department of Physiology, School of Preclinical Medicine, Chengdu University, Chengdu, Sichuan Province, People's Republic of China

* yepeng@cdu.edu.cn (PY); weiyuanyuan@cdu.edu.cn (YW)

## Abstract

## Introduction

Before anti-EGFR therapy is given to patients with colorectal cancer, it is required to determine *KRAS* mutation status in tumor. When tumor tissue is not available, cell-free DNA (liquid biopsy) is commonly used as an alternative. Due to the low abundance of tumor-derived DNA in cell-free DNA samples, methods with high sensitivity were preferred, including digital polymerase chain reaction, amplification refractory mutation system and next-generation sequencing. The aim of this systemic review and meta-analysis was to investigate the accuracy of those methods in detecting *KRAS* mutation in cell-free DNA sample from patients with colorectal cancer.

## Methods

Literature search was performed in Pubmed, Embase, and Cochrane Library. After removing duplicates from the 170 publications found by literature search, eligible studies were identified using pre-defined criteria. Quality of the publications and relevant data were assessed and extracted thereafter. Meta-DiSc and STATA softwares were used to pool the accuracy parameters from the extracted data.

## Results

A total of 33 eligible studies were identified for this systemic review and meta-analysis. After pooling, the overall sensitivity, specificity, and diagnostic odds ratio were 0.77 (95%CI: 0.74–0.79), 0.87 (95%CI: 0.85–0.89), and 23.96 (95%CI: 13.72–41.84), respectively. The overall positive and negative likelihood ratios were 5.55 (95%CI: 3.76–8.19) and 0.29 (95% CI: 0.21–0.38), respectively. Area under curve of the summarized ROC curve was 0.8992.

**Data Availability Statement:** All data files are available from the Systematic Review Data

Repository (SRDR) database (accession number 1639, URL: https://srdr.ahrq.gov/projects/1639).

**Funding:** This work was supported by National Natural Science Foundation of China (No.: 81160546; http://www.nsfc.gov.cn/english/site_1/index.html) to YW. The funders had no role in study design, data collection and analysis, decision to publish, or preparation of the manuscript.

**Competing interests:** The authors have declared that no competing interests exist.

## Conclusion

Digital polymerase chain reaction, amplification refractory mutation system, and next-generation sequencing had overall high accuracy in detecting *KRAS* mutation in cell-free DNA sample. Large prospective randomized clinical trials are needed to further convince the accuracy and usefulness of *KRAS* mutation detection using cfDNA/liquid biopsy samples in clinical practice.

## Trial registration

PROSPERO CRD42020176682; https://www.crd.york.ac.uk/prospero/display_record.php?RecordID=176682.

## Introduction

Colorectal cancer (CRC) is currently a leading cause of cancer-related death worldwide [1]. For resectable CRC, surgery remains the standard of care, while for non-resectable tumors, patients are mostly treated by chemotherapy and targeted therapy, e.g. anti-epithelial growth factor receptor (EGFR) therapy [2,3]. As examples of anti-EGFR therapy, cetuximab and panitumumab were firstly approved for the treatment of chemorefractory metastatic CRC (mCRC) in 2004 and 2006, respectively [2]. In subsequent investigations of the two drugs as second-line treatment of mCRC, several large phase III clinical trials showed benefit in response rate and progression-free survival, but not in overall survival [4–6]. Retrospective analysis on the results of those trials revealed that mCRC patients with different molecular background differed significantly in treatment response. The first finding was that mutations in *KRAS* exon 2 were linked to poor response in mCRC patients treated by anti-EGFR therapy [7]. Later, mutations in *KRAS* exon 3 and 4 (codons 61, 117, and 146), and in *NRAS* exons 2, 3, 4 were also found to be associated with resistance to anti-EGFR therapy in mCRC patients [8,9]. Retrospective analysis showed that patients with *RAS* wild-type tumors had significant benefit in all efficacy end points, while no significant benefit was observed in patients with *RAS* mutated tumors [10]. Those compelling evidences led the administrations to limit the use of cetuximab and panitumumab only in mCRC patients with both *KRAS* wild-type and *NRAS* wild-type, and require the test of *KRAS* and *NRAS* mutation status before anti-EGFR treatment is given [2,11].

The detection of *KRAS* and *NRAS* mutation in CRC is mostly performed on archived surgical tumor tissue samples or tumor biopsy samples using traditional point mutation detection methods (e.g. quantitative polymerase chain reaction, quantitative PCR and amplification refractory mutation system, ARMS) or sequencing technique (e.g. Sanger sequencing) [12–14]. However, for refractory CRC or mCRC patients, tumor tissue samples are often not available. A small proportion of cell-free DNA (cfDNA) in liquid biopsy sample (plasma, urine, and etc.) derives from tumor cells (also called circulating tumor DNA, ctDNA), and could serve as an alternative source of tumor-derived DNA to surgical tumor tissue sample or tumor biopsy sample [15,16].

The liquid biopsy-based tumor genotyping has been intensively studied in recent years [17]. Due to the low abundance of ctDNA, several high-sensitivity techniques have been developed and evaluated for the tumor genotyping in liquid biopsy samples, including digital PCR, ARMS, and next generation sequencing (NGS) [18–20]. ARMS is already commonly used in clinical laboratories, with acceptable accuracy and low cost [21]. The high-throughput

technology, NGS, has the ability to detect hundreds of mutations in a run, but is challenged by its relatively low sensitivity and high cost [22]. Digital PCR is well known by its high sensitivity, but the cost of this technique is still higher than traditional quantitative PCR [23]. For the detection of *KRAS* mutations, the limit of detection for ARMS, NGS, and digital PCR was reported to be 1%, 2–6%, or as low as 0.01%, respectively [12,22,24,25]. However, although the limit of detection of those techniques was determined, their performance in clinical practice has not been fully validated yet. The aim of this systematic review and meta-analysis was to investigate the accuracy of *KRAS* mutation detection in cfDNA samples from patients with CRC, compared to paired tissue samples. After searching of eligible studies in databases, and subsequent extraction and analysis of data, those techniques (digital PCR, ARMS, and NGS) showed overall high accuracy in detecting *KRAS* mutation in cfDNA samples of colorectal cancer patients.

## Materials and methods

### Registration and publication of study protocol

Study protocol of this systemic review and meta-analysis has been registered on International prospective register of systemic reviews (PROSPERO) and the registration number is CRD42020176682. Detailed study protocol has been published [26].

### Literature searching and selection of publication

Literature research was performed independently by PY and PC in June 2020, and no limitation was placed on publication date. Pubmed, Embase, and Cochrane Library were searched using "KRAS", "digital PCR", "next-generation sequencing", "amplification refractory mutation system", "cell-free DNA", "circulating tumor DNA", "liquid biopsy", and "colorectal cancer". Alternative spelling or abbreviations were also included in the search (see S1 Table for detailed search strategy). In the searching results, we firstly reviewed the titles and abstracts of the publications. Duplicated publications were removed and irrelevant publications were excluded using the following criteria: 1) not a human study; 2) not describing *KRAS* mutation; 3) no liquid biopsy samples or tissue samples included; 4) did not use any techniques among digital PCR, ARMS, and NGS; 5) not colorectal cancer; 6) reviews, abstracts, letter to the editor, comments, case reports, or studies with un-interpretable data.

For the remaining publications, full texts were downloaded and examined. Studies were further excluded if 1) data were un-interpretable or mixed (cannot be separated from results of other gene mutations); 2) lacking *KRAS* mutated or *KRAS* wild-type tissue samples; 3) no defined criteria for the positively/negativity of *KRAS* mutation. In the resulting eligible studies, *KRAS* testing results from paired cfDNA samples and tumor tissue samples (*KRAS* mutated or wild-type) were extracted from each publication, including sample size, and numbers of true positive, false positive, false negative, and true negative. Sample types of cfDNA (e.g. plasma, serum, etc.) and techniques used for cfDNA samples or tissue samples were also extracted from each of the eligible studies. If multiple techniques were used to determine *KRAS* mutation status in the same patient cohort, one technique was selected for data extraction using the following criteria: 1) technique used for a larger number of samples; 2) technique having similar detection region with the technique used for *KRAS* detection in paired tissue samples. Other information was also extracted and recorded, including age of patients, race of patients (Caucasian, Asian, etc.), country of origin (region of the study), type of CRC (metastatic or non-metastatic), and name of the first author of the publication. Disagreement in the literature search results between the two researchers (PY and PC) was solved by a third researcher (YW).

Each of the eligible studies included in the data extraction was evaluated using quality assessment of diagnostic accuracy studies 2 (QUADAS-2) [27].

## Statistical analysis

Sensitivity, specificity, positive likelihood ratio (PLR), negative likelihood ratio (NLR), and diagnostic odds ratio (DOR) of the eligible studies were pooled using Meta-DiSc statistical software version 1.4 [28], and summary receiver operating characteristic (SROC) and area under curve (AUC) were also generated. Cochran-Q and $I^2$ were used to evaluate inter-study heterogeneity. Random effects model (DerSimonian-Laird) was used for pooling the results if significant heterogeneity was observed ($I^2 \geq 50\%$ and $P \leq 0.05$), while fixed effects model (Mantel-Haenszel) was used if no significant heterogeneity was identified. Threshold analysis and meta-regression were performed using Meta DiSc to search for potential source of heterogeneity. Publication bias was evaluated using Deek's funnel plot asymmetry test performed by STATA 12.0 (STATA Corp.). $P < 0.05$ was considered statistically significant.

## Results

### Search results

After literature searching, a total of 170 publications were found from Pubmed (73 publications), Embase (77 publications), and Cochrane Library (20 publications), as shown in Fig 1. After removal of duplicated (62 publications) and irrelevant publications (53 publications), full text of the rest 55 publications were reviewed and another 22 studies were excluded due to lack of *KRAS* mutated/wild-type tissue samples or un-interpretable data. Data were extracted from the rest 33 eligible studies and meta-analysis was performed.

### Review of eligible publications

As shown in Table 1, in the 33 eligible studies, *KRAS* status in cfDNA samples was tested using NGS in 15 studies, digital PCR in 17 studies, or ARMS in 1 study. All the eligible studies used from plasma samples, except for the study by Kitagawa et al [14] which used serum instead. Due to the emergence of the All-RAS sequencing concept, many of the studies tested both *KRAS* and *NRAS* (and even *HRAS*), as well as the expanded isoforms of those genes [19,29–47]. Most of those studies reported separate results for *KRAS*, and the rest 6 studies reported All-*RAS* status [30,40,41,44–46]. For those 6 studies, *NRAS* results were also included in the subsequent systematic review and meta-analysis. The accuracy of *KRAS*/All-*RAS* status detection in cfDNA samples in each study is summarized below.

**NGS.** In the 15 studies using NGS to measure *KRAS* status in cfDNA samples, studies by Kato [19], Kim [32], Gupta [35], or Choi [36] used commercial Guardant360 NGS panel (Guardant Health Inc.), and results showed sensitivity of 59.5%, 83.3%, 75.9%, or 63.6%, and specificity of 87.2%, 86.9%, 97.8%, or 92.9%, respectively. Three of those 4 studies used Foundation One NGS panel (Foundation Medicine) to test *KRAS* mutation status in paired tissue samples [19,35,36], while the other study by Kim [32] used traditional Sanger sequencing instead. Studies by Osumi [37], or Rachiglio [46] used commercial Oncomine™ NGS panels (Life Technologies, covering 14 genes or 22 genes, respectively) to detect *KRAS* or All-*RAS* status in cfDNA samples of CRC patients, and the sensitivity was 80.6% or 63.2%, and specificity was 81.5% or 100%, respectively. Beranek et al [47] used another commercial NGS panel (Somatic 1, Multiplicom, Belgium) in a 12-patient cohort. The concordance rate was 86% between cfDNA and tumor tissue samples, and the calculated sensitivity and specificity were 80% and 100%, respectively.

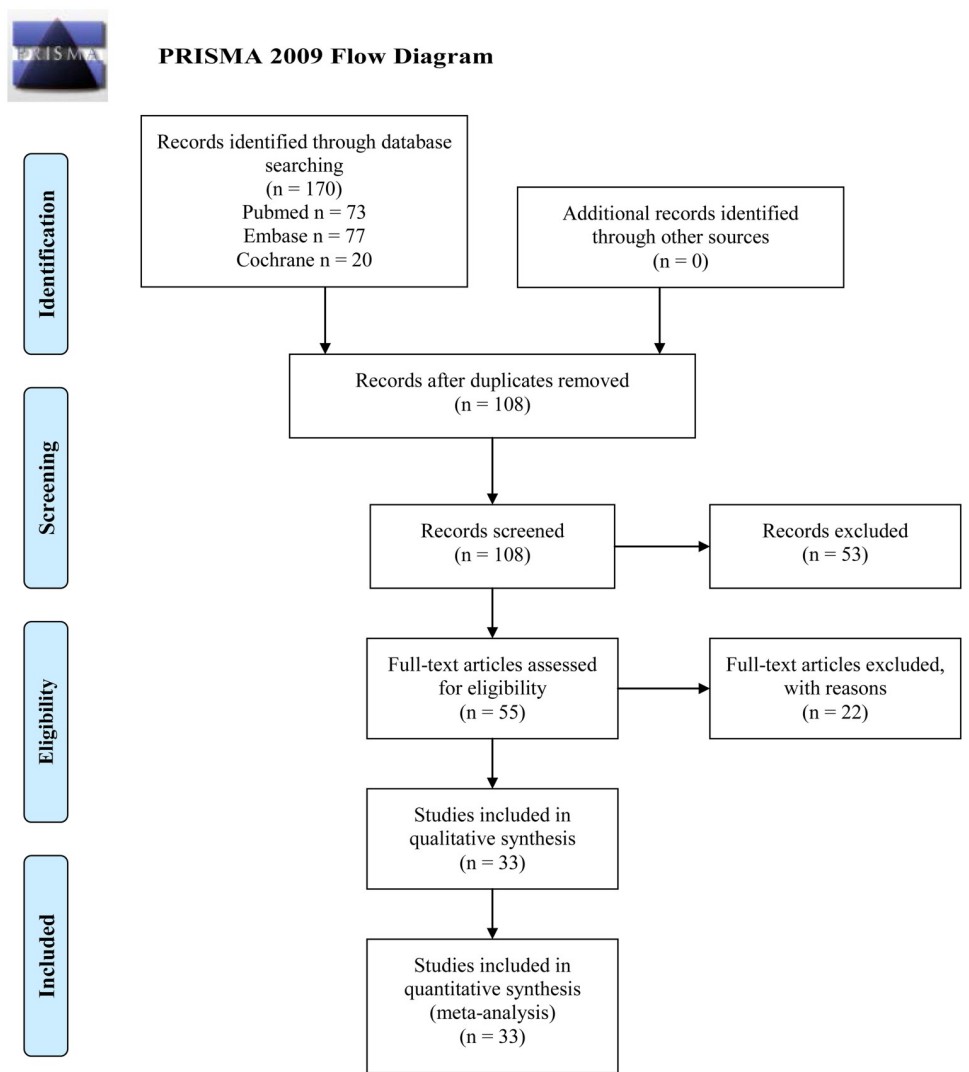

**Fig 1. PRISMA 2009 flow diagram.** *From*: Moher D, Liberati A, Tetzlaff J, Altman DG, The PRISMA Group (2009). *P*referred *R*eporting *I*tems for Systematic Reviews and *M*eta-*A*nalyses: The PRISMA Statement. PLoS Med 6(7): e1000097. doi:10.1371/journal.pmed1000097 **For more information, visit** www.prisma-statement.org.

The rest 8 studies all used customized NGS panels for cfDNA samples. Kang et al [13] performed a liquid-biopsy-based tumor profiling using a 10-gene NGS panel in 48 mCRC patients, and the calculated sensitivity and specificity were 86.4% and 65.4%. Chang et al [29] analyzed correlation of genomic alterations between paired tumor tissue and plasma samples using a 275-gene NGS panel in 21 patients with different cancer types. In the 5 patients with CRC, the calculated concordance rate of *KRAS* status was 60% (3/5), with 2 false-negative cases. Wang et al [31] investigated the *KRAS* mutation status in paired plasma and tumor tissue samples of 97 mCRC patients using NGS, and results showed concordance rate of 65.26%, sensitivity of 70.02%, and specificity of 66.71%. In the study by Kidess-Sigal et al [50], the authors compared *KRAS*, *BRAF*, and *PIK3CA* status between circulating tumor cells, ctDNA, and tissue samples of 15 mCRC patients using a 4-gene NGS panel, and in the 3 patients with both ctDNA and primary tumor samples available, the calculated concordance rate for *KRAS*

**Table 1. Summary of studies comparing *KRAS*/All-*RAS* mutation status in cfDNA and tumor tissue samples from colorectal cancer patients.**

| Author, year | Sample size | Technique used for cfDNA samples | Sample type for cfDNA | Technique used for tissue samples | Region of the study | Type of colorectal caner |
|---|---|---|---|---|---|---|
| **Bidard et al, 2019** [48] | 125 | digital droplet PCR (Bio-Rad) | Plasma | standard routine technique | Europe | metastatic |
| **Sclafani et al, 2018** [49] | 90 | digital droplet PCR (Bio-Rad) | Plasma | PCR & Sanger sequencing | Europe | primary |
| **Kang et al, 2020** [13] | 48 | NGS (customized panel) | Plasma | Sanger sequencing | Asia | metastatic |
| **Kato et al, 2019** [19] | 76 | NGS (Guardant360, Guardant Health) | Plasma | NGS (Foundation One, Foundation Medicine) | America | either primary or metastatic |
| **Chang et al, 2018** [29] | 5 | NGS (275-gene panel from Qiagen) | Plasma | NGS (275-gene panel from Qiagen) | Asia | either primary or metastatic |
| **Garcia et al, 2018** [30] | 28 | Beads, Emulsion, Amplification and Magnetics (BEAMing) (OncoBEAM™-RAS-CRC kit[a]) | Plasma | NGS (customized Ampliseq library,) | Europe | metastatic |
| **Wang et al, 2017** [31] | 97 | NGS (commercial panel from SinoMD) | Plasma | standard routine technique | Asia | metastatic |
| **Kidess-Sigal et al, 2016** [50] | 3 | NGS (SCODA[b] mutation enrichment and detection technology) | Plasma | Sanger sequencing | America | metastatic |
| **Kim et al, 2015** [32] | 29 | NGS (Guardant360, Guardant Health) | Plasma | Sanger sequencing | Asia | metastatic |
| **Vessies et al, 2020** [33] | 6 | BEAMing (OncoBEAM™-RAS-CRC kit[a]) | Plasma | standard of care | Europe | metastatic |
| **Cao et al, 2020** [34] | 35 | NGS (customized 605-gene panel) | Plasma | NGS (whole exome sequencing) | Asia | either primary or metastatic |
| **Gupta et al, 2020** [35] | 75 | NGS (Guardant360, Guardant Health) | Plasma | NGS (Foundation One, Foundation Medicine) | America | metastatic |
| **Kitagawa et al, 2019** [14] | 40 | digital droplet PCR (Bio-Rad) | Serum | ARMS or Luminex | Asia | either primary or metastatic |
| **Galbiati et al, 2019** [51] | 20 | digital droplet PCR (Bio-Rad) | Plasma | MassARRAY (Sequenom) | Europe | metastatic |
| **Choi et al, 2019** [36] | 61 | NGS (Guardant360, Guardant Health) | Plasma | NGS (Foundation One, Foundation Medicine) | America | either primary or metastatic |
| **Liebs, et al, 2019** [18] | 53 | digital droplet PCR (Bio-Rad) | Plasma | digital droplet PCR (Bio-Rad) | Europe | either primary or metastatic |
| **Osumi et al, 2019** [37] | 101 | NGS (Oncomine™ Colon cfDNA assay) | Plasma | PCR (RASKET KIT, Luminex) | Asia | metastatic |
| **Galbiati et al, 2019** [52] | 30 | digital droplet PCR (Bio-Rad) | Plasma | MassARRAY (Sequenom) | Europe | metastatic |
| **Takayama et al, 2018** [53] | 85 | digital droplet PCR (Bio-Rad) | Plasma | ARMS/PCR (RASKET kit) | Asia | metastatic |
| **Yao et al, 2018** [38] | 64 | NGS (Sureselect, Agilent, targeting *KRAS/NRAS/HRAS/BRAF*) | Plasma | ARMS (Human KRAS/NRAS/BRAF mutations detection kit[c]) | Asia | metastatic |
| **Sun et al, 2018** [39] | 11 | NGS (customized 85-gene colorectal cancer gene panel) | Plasma | NGS (customized 85-gene colorectal cancer gene panel) | Asia | either primary or metastatic |
| **Normanno et al, 2018** [40] | 92 | BEAMing (OncoBEAM™-RAS-CRC kit[a]) | Plasma | NGS (AmpliSeq Colon and Lung Cancer Panel, ThermoFisher) | Europe | metastatic |
| **Garcia-Foncillas et al, 2018** [41] | 236 | BEAMing (OncoBEAM™-RAS-CRC kit[a]) | Plasma | standard of care (Pyrosequencing/ Cobas/Therascreen/Idylla/CLART-CMA kit) | Europe | metastatic |
| **Beije et al, 2016** [42] | 12 | NGS (customized 21-gene CRC-specific panel) | Plasma | NGS (customized 21-gene CRC-specific panel) | Europe | metastatic |
| **Sefrioui et al, 2017** [54] | 29 | QuantStudio™ 3D digital PCR | Plasma | SNaPshot multiplex assay | Europe | metastatic |

(*Continued*)

**Table 1.** (Continued)

| Author, year | Sample size | Technique used for cfDNA samples | Sample type for cfDNA | Technique used for tissue samples | Region of the study | Type of colorectal caner |
|---|---|---|---|---|---|---|
| Grasselli et al, 2017 [43] | 117 | BEAMing (OncoBEAM™-RAS-CRC kit[a]) | Plasma | BEAMing (OncoBEAM™-RAS-CRC kit[a]) | Europe | metastatic |
| Schmiegel et al, 2017 [44] | 98 | BEAMing (OncoBEAM™-RAS-CRC kit[a]) | Plasma | standard of care (Pyrosequencing/Sanger sequencing/NGS) | Europe | metastatic |
| Vidal et al, 2017 [45] | 115 | BEAMing (OncoBEAM™-RAS-CRC kit[a]) | Plasma | standard of care[d] | Europe | metastatic |
| Beranek et al, 2016 [47] | 32 | NGS (Somatic 1[e]) | Plasma | Standard routine technique | Europe | metastatic |
| Rachiglio et al, 2016 [46] | 35 | NGS (22-gene Oncomine™ Solid Tumor DNA kit, Life Technologies) | Plasma | Pyrosequencing (Therascreen KRAS and NRAS Pyro kit, Qiagen) | Europe | metastatic |
| Yamada et al, 2016 [55] | 94 | QuantStudio™ 3D digital PCR | Plasma | MEBGEN-Luminex method | Asia | metastatic |
| Spindler et al, 2015 [20] | 211 | ARMS-based in-house assay | Plasma | ARMS-based in-house assay | Europe | metastatic |
| Taly et al, 2013 [12] | 50 | picodroplet digital PCR | Plasma | quantitative PCR | Europe | metastatic |

[a]From Sysmex Inostics, which targets 34 variants in *KRAS* and *NRAS* genes.

[b]Sequence-specific synchronous coefficient of drag alteration.

[c]From Beijing ACCB Biotech, which targets *KRAS/NRAS* codons 12, 13, 59, 61, 117, 146, and *BRAF* codon 600.

[d]Therascreen KRAS RGQ PCR kit (Qiagen), COBAS KRAS mutation test (Roche), or pyrosequencing (PyroMark, Qiagen).

[e]From Multiplicom, Belgium, which targets *BRAF*, *KRAS*, and *NRAS*.

was 66.7% (2/3), with 1 false-positive case. In the study by Cao et al [34], a 605-gene NGS panel was used to analyze tumor and plasma samples of CRC patients, and from 35 patients with *KRAS* status available in both tumor and plasma, the calculated sensitivity was 75.0% and specificity was 82.6%. Using a customized targeted NGS library kit (SureSelect QXT, Agilent Technologies, USA), Yao et al [38] investigated *KRAS/NRAF/BRAF* mutations in plasma and tumor samples of mCRC patients, and concordance rate of *KRAS* was 81.25% in 64 patients, with sensitivity of 66.7% and specificity of 90.0%. Sun et al [39] used an 85-gene NGS panel and analyzed paired tumor and plasma samples of 11 CRC patients, and results showed calculated sensitivity of 80% and specificity of 100% in the detection of *KRAS* status in cfDNA. Beije et al [42] designed a CRC-specific 21-gene NGS panel and plasma samples from 12 mCRC patients analyzed by this panel showed calculated sensitivity of 50% and specificity of 87.5% in *KRAS* mutation detection.

**Digital PCR.** Digital PCR was used to detect *KRAS* status in cfDNA samples in 17 of the eligible studies. Within those studies, 7 publications by Bidard [48], Sclafani [49], Kitagawa [14], Galbiati [51,52], Liebs [18], or Takayama [53] used digital droplet PCR (Bio-Rad) targeting *KRAS* point mutations in cfDNA samples, and the calculated sensitivity of this technique ranged from 36% to 100% (91.3%, 42.9%, 94.1%, 100%, 66.7%, 36%, or 79.3%, respectively) and specificity ranged from 50% to 100% (92.4%, 64.5%, 100%, 92.3%, 66.7%, 100%, 50%, respectively).

Seven studies by Garcia [30], Vessies [33], Normanno [40], Grasselli [43], Schmiegel [44], Vidal [45], or García-Foncillas [41] used BEAMing digital PCR technology. All those studies used a commercial OncoBEAM™ RAS CRC Kit (Sysmex Inostics) which could target 34 somatic mutations in *KRAS/NRAS exons 2*, *3*, *4* in one run. Study by Garcia et al [30]

compared the accuracy of different platforms, including BEAMing which showed sensitivity of 93.3% and specificity of 69.2% in detecting mutations in *KRAS/NRAS* in plasma samples, compared to tissue sample results. From a small patient cohort (6 patients), Vessies et al [33] investigated the performance of 4 platforms and BEAMing platform correctly identified *KRAS* mutations in plasma samples of 4 patients, with 1 false positive and 1 false negative cases. Normanno et al [40] analyzed All-*RAS* mutations in plasma samples from a sub-cohort of patients from a clinical trial (CAPRI-GOIM study) using BEAMing and digital droplet PCR, and the results revealed a sensitivity of 70.0% and specificity of 83.1% of BEAMing platform. Grasselli et al [43] also investigated the performance of BEAMing in detecting *KRAS* mutation in plasma and tissue of metastatic colorectal cancer and results showed sensitivity of 85.7%, specificity of 94%, and concordance rate of 89.7%. The rest 3 studies by Schmiegel [44], Vidal [45], or García-Foncillas [41] investigated All-*RAS* status in plasma samples from mCRC patients using BEAMing and compared the results with paired tissue samples. The results showed concordance rate of 91.8%, 93%, or 89%, sensitivity of 90.4%, 96.4%, or 86.3%, and specificity of 93.5%, 90%, or 92.4%, respectively.

In the rest 3 publications, 2 studies by Sefrioui [54] or Yamada [55] used chip-based digital PCR platform (QuantStudio™ 3D Digital PCR System, Thermo Fisher Scientific) in the detection of *KRAS* status in plasma, and obtained sensitivity of 85.7% or 79.5%, and specificity of 100% or 90.9%, respectively. Taly et al [12] used picodroplet digital PCR technique in detecting *KRAS* mutation in plasma samples from 50 mCRC patients, and results showed calculated sensitivity of 73.7% and specificity of 93.5%.

**ARMS.**   Only 1 study used ARMS to detect *KRAS* status in cfDNA samples. Spindler et al [20] used ARMS method to detect *KRAS* mutation in matched tumor tissue and plasma samples from 211 mCRC patients, and the overall concordance rate was 85.0%, with sensitivity of 80.0% and specificity of 95.8%.

In conclusion, the 33 studies comprised 2203 CRC patients with paired cfDNA and tumor tissue samples. Out of the 33 eligible studies, 20 showed high concordance (higher than 80%) in *KRAS* detection results between cfDNA and tumor tissue samples. High specificity (higher than 80%) was also observed in majority (26 out of 29) of the studies. More than half (17 out of 33) of the studies showed high sensitivity (higher than 80%).

## Quality assessment of eligible studies

QUADAS-2 was used to assess the quality of individual studies and the result is shown in Table 2. In the four aspects of risk of bias assessment, percentage of high risk of bias was from 0% (n = 0, patient selection, reference standard) to 15% (n = 5, flow and timing). Percentage of low risk of bias was from 24% (n = 8, index test, flow and timing) to 73% (n = 24, patient selection). Flow and timing showed the highest risk of bias (15% high risk of bias and 24% low risk of bias). Patient selection showed the lowest risk of bias (0% high risk of bias and 73% low risk of bias). For the applicability of studies, all the studies were classified as low concern in the three aspects (patient selection, index test, and reference standard).

## Meta-analysis of the accuracy of KRAS mutation detection using cfDNA samples

The *KRAS* detection results of the 2203 CRC patients were pooled using statistical software. As shown in Fig 2, results showed a pooled sensitivity of 0.77 [95% confidence interval (CI): 0.74–0.79] and pooled specificity of 0.87 (95%CI: 0.85–0.89). The pooled PLR, NLR and DOR were 5.55 (95%CI: 3.76–8.19), 0.29 (95%CI: 0.21–0.38), and 23.96 (95%CI: 13.72–41.84), respectively. SROC curve was also generated and the AUC was 0.8992.

**Table 2. QUADAS-2 assessment of eligible studies.**

| Author, year | Risk of bias | | | | Applicability concerns | | |
|---|---|---|---|---|---|---|---|
| | Patient selection | Index test | Reference standard | Flow and timing | Patient selection | Index test | Reference standard |
| Bidard et al, 2019 [48] | low | low | low | unclear | low | low | low |
| Sclafani et al, 2018 [49] | low | unclear | low | high | low | low | low |
| Kang et al, 2020 [13] | low | unclear | unclear | high | low | low | low |
| Kato et al, 2019 [19] | low | unclear | unclear | high | low | low | low |
| Chang et al, 2018 [29] | unclear | high | low | unclear | low | low | low |
| Garcia et al, 2018 [30] | low | low | low | low | low | low | low |
| Wang et al, 2017 [31] | unclear | unclear | low | unclear | low | low | low |
| Kidess-Sigal et al, 2016 [50] | low | unclear | unclear | unclear | low | low | low |
| Kim et al, 2015 [32] | low | unclear | low | unclear | low | low | low |
| Vessies et al, 2020 [33] | low | unclear | low | high | low | low | low |
| Cao et al, 2020 [34] | low | unclear | unclear | unclear | low | low | low |
| Gupta et al, 2020 [35] | low | low | low | unclear | low | low | low |
| Kitagawa et al, 2019 [14] | unclear | unclear | unclear | unclear | low | low | low |
| Galbiati et al, 2019 [51] | low | unclear | low | unclear | low | low | low |
| Choi et al, 2019 [36] | low | low | low | unclear | low | low | low |
| Liebs, et al, 2019 [18] | unclear | unclear | unclear | unclear | low | low | low |
| Osumi et al, 2019 [37] | low | unclear | unclear | unclear | low | low | low |
| Galbiati et al, 2019 [52] | unclear | unclear | low | low | low | low | low |
| Takayama et al, 2018 [53] | unclear | unclear | low | unclear | low | low | low |
| Yao et al, 2018 [38] | low | unclear | low | low | low | low | low |
| Sun et al, 2018 [39] | low | low | low | low | low | low | low |
| Normanno et al, 2018 [40] | low | unclear | unclear | unclear | low | low | low |
| García-Foncillas et al, 2018 [41] | unclear | unclear | low | unclear | low | low | low |
| Beije et al, 2016 [42] | low | low | low | low | low | low | low |
| Sefrioui et al, 2017 [54] | low | unclear | low | high | low | low | low |
| Grasselli et al, 2017 [43] | low | low | low | unclear | low | low | low |
| Schmiegel et al, 2017 [44] | low | unclear | unclear | low | low | low | low |
| Vidal et al, 2017 [45] | low | unclear | low | unclear | low | low | low |
| Beranek et al, 2016 [47] | unclear | unclear | unclear | unclear | low | low | low |
| Rachiglio et al, 2016 [46] | low | unclear | unclear | low | low | low | low |
| Yamada et al, 2016 [55] | low | low | low | low | low | low | low |
| Spindler et al, 2015 [20] | low | unclear | unclear | unclear | low | low | low |
| Taly et al, 2013 [12] | unclear | unclear | unclear | unclear | low | low | low |

All the forest plots of the meta-analysis (see Fig 2) showed significant inter-study heterogeneity ($I^2 \geq 50\%$ and $P \leq .05$), indicating significant differences among the studies. Therefore, we focused more on the possible sources of inter-study heterogeneity and subgroup analysis. The Spearman correlation coefficient was -0.053 ($P = 0.77$), suggesting no significant threshold effect. In the meta-regression analysis, we included 5 covariates (technique used for cfDNA samples, technique used for tissue samples, region, race, type of CRC), and results indicated that inter-study heterogeneity was not associated to technique used for cfDNA samples ($P = 0.85$), technique used for tissue samples ($P = 0.22$), region ($P = 0.91$), race of patients ($P = 0.68$), and type of CRC ($P = 0.20$). Age of patients was excluded from the meta-regression because clear age data were not provided in several studies [33,51,52,54,55]. Type of liquid biopsy samples was also not included as a covariate in the meta-regression since almost all the eligible studies used plasma samples (only 1 study used serum samples instead [14]).

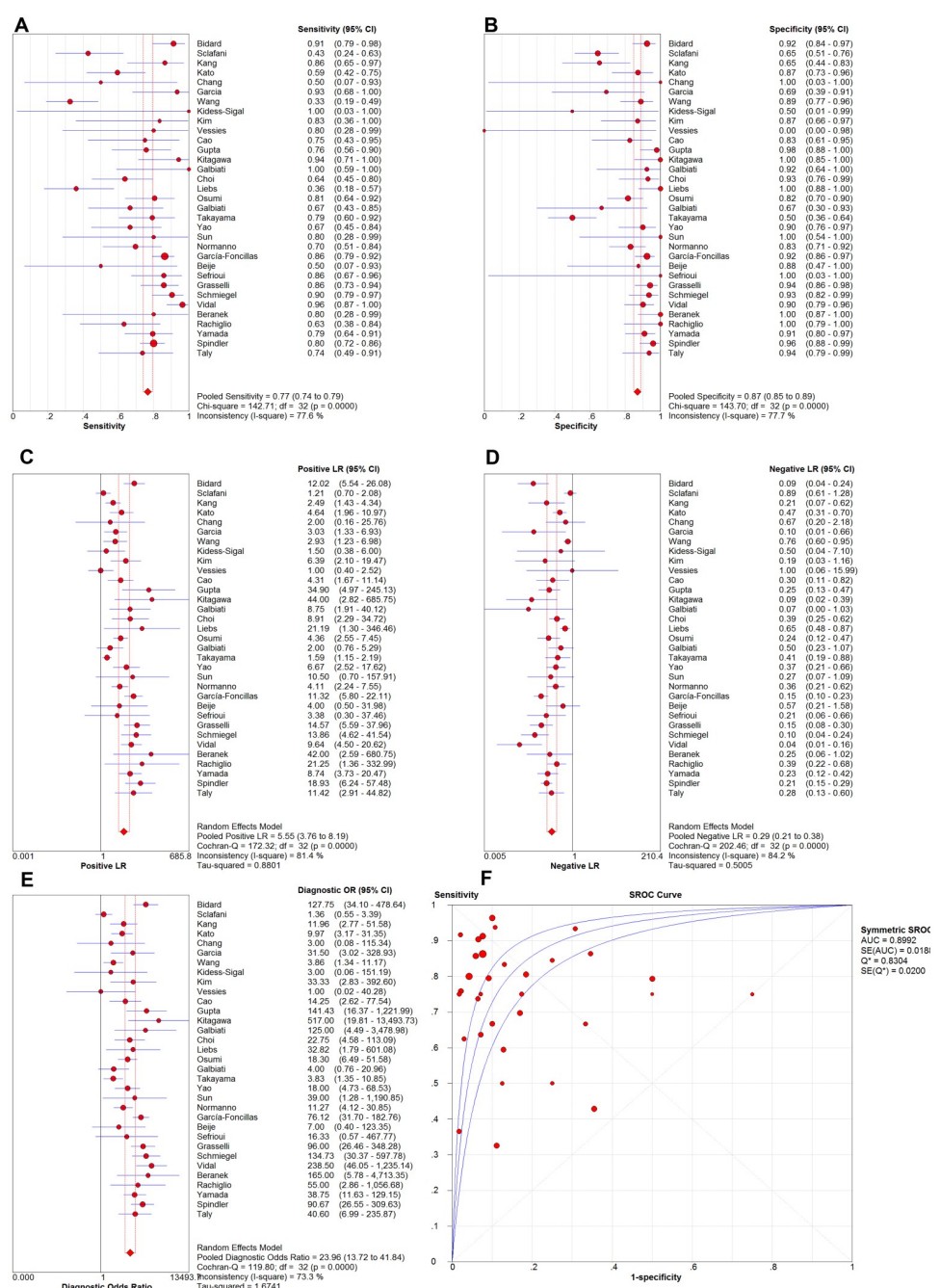

**Fig 2. Pooled sensitivity, specificity, PLR, NLR, DOR and SROC of the eligible studies.**

Subgroup analysis was conducted according to type of CRC. Several studies involved in this systemic review and meta-analysis attempted to investigate possible roles of liquid biopsy in early detection or monitoring of the disease (e.g. driver mutations, resistance to targeted therapeutics), and therefore included CRC patients of different stages (I-IV) [14,18,19,29,34,36,39]. Since both primary and metastatic CRC patients were involved in those studies [14,18,19,39], we extracted their data separately. Three studies were excluded from the subgroup analysis

**Table 3. Meta-analysis results.**

| | No. of studies/patient cohorts | Sensitivity | Specificity | PLR | NLR | DOR | AUC of SROC |
|---|---|---|---|---|---|---|---|
| **Overall** | 33 | 0.77(0.74–0.79) | 0.87(0.85–0.89) | 5.55(3.76–8.19) | 0.29(0.21–0.38) | 23.96(13.72–41.84) | 0.8992 |
| **Type of CRC** | | | | | | | |
| **metastatic** | 24[a] | 0.79(0.76–0.82) | 0.88(0.86–0.90) | 5.14(3.32–7.98) | 0.26(0.19–0.35) | 29.17(17.00–50.06) | 0.9045 |
| **primary** | 4[b] | 0.57(0.41–0.72) | 0.73(0.62–0.82) | 2.08(1.25–3.44) | 0.46(0.20–1.07) | 10.81(1.00–117.04) | 0.7304 |
| **Technique used for cfDNA samples** | | | | | | | |
| **NGS** | 15 | 0.65(0.59–0.71) | 0.88(0.85–0.91) | 5.21(3.97–6.83) | 0.38(0.28–0.52) | 14.61(9.78–21.84) | 0.8574 |
| **Digital PCR** | 17 | 0.81(0.78–0.85) | 0.85(0.83–0.88) | 5.51(3.02–10.07) | 0.23(0.14–0.37) | 29.18(11.79–72.25) | 0.9067 |
| **NGS (Guardant360)** | 4 | 0.67(0.57–0.76) | 0.92(0.86–0.96) | 8.49(4.66–15.46) | 0.35(0.26–0.47) | 22.47(10.58–47.75) | 0.8260 |
| **BEAMing** | 7 | 0.87(0.83–0.90) | 0.90(0.86–0.93) | 5.94(2.86–12.34) | 0.15(0.09–0.27) | 50.96(18.56–139.92) | 0.9388 |
| **Digital droplet PCR** | 7 | 0.71(0.64–0.78) | 0.78(0.72–0.83) | 4.37(1.66–11.53) | 0.33(0.16–0.69) | 18.20(3.45–96.00) | 0.8703 |
| **Region of the study** | | | | | | | |
| **Europe** | 18 | 0.80(0.77–0.83) | 0.89(0.87–0.92) | 6.42(3.63–11.35) | 0.25(0.16–0.39) | 32.97(13.63–79.79) | 0.9143 |
| **Asia** | 11 | 0.71(0.65–0.77) | 0.81(0.77–0.85) | 4.19(2.48–7.07) | 0.31(0.19–0.53) | 13.87(7–27.48) | 0.8539 |
| **America** | 4 | 0.66(0.56–0.75) | 0.92(0.86–0.96) | 5.99(1.87–19.15) | 0.37(0.28–0.49) | 19.98(9.26–43.13) | 0.6545 |
| **Region of studies using NGS for cfDNA samples** | | | | | | | |
| **Europe** | 3 | 0.64(0.44–0.81) | 0.98(0.90–1.00) | 15.59(3.57–68.07) | 0.37(0.23–0.62) | 35.79(6.76–189.46) | 0.1279 |
| **Asia** | 8 | 0.64(0.56–0.72) | 0.84(0.79–0.88) | 3.95(2.92–5.33) | 0.36(0.20–0.65) | 11.21(6.81–18.46) | 0.8683 |
| **America** | 4 | 0.66(0.56–0.75) | 0.92(0.86–0.96) | 5.99(1.87–19.15) | 0.37(0.28–0.49) | 19.98(9.26–43.13) | 0.6545 |
| **Region of studies using digital PCR for cfDNA samples** | | | | | | | |
| **Europe** | 14 | 0.81(0.78–0.85) | 0.88(0.85–0.90) | 5.48(2.98–10.05) | 0.23(0.12–0.41) | 29.63(10.54–83.32) | 0.9119 |
| **Asia** | 3 | 0.82(0.73–0.90) | 0.75(0.67–0.82) | 6.59(0.78–55.14) | 0.23(0.15–0.37) | 28.33(2.87–279.59) | 0.8758 |
| **Subtypes of digital PCR in studies from Europe** | | | | | | | |
| **BEAMing** | 7 | 0.87(0.83–0.90) | 0.90(0.86–0.93) | 5.94(2.86–12.34) | 0.15(0.09–0.27) | 50.96(18.56–139.92) | 0.9388 |
| **Digital droplet PCR** | 5 | 0.66(0.57–0.74) | 0.83(0.77–0.88) | 4.63(1.31–16.40) | 0.38(0.17–0.89) | 16.70(1.77–157.26) | 0.8750 |

[a]One study/patient cohort [39] was excluded by statistical software due to lack of true positive samples.

[b]One study/patient cohort [18] was excluded by statistical software due to lack of true positive samples.

because we cannot separate their data by groups of primary and metastatic CRC [29,34,36]. As shown in Table 3, compared to primary CRC, mCRC showed higher pooled sensitivity [0.79 (95%CI: 0.76–0.82)] and specificity [0.88 (95%CI: 0.86–0.90)]. The pooled DOR [29.17 (95% CI: 17.00–50.06)] and AUC of SROC curve (0.9045) of mCRC were also higher than that of primary CRC [10.81 (95%CI: 1.00–117.04) and 0.7304]. For the comparison between early- (I, II) and late-stage (III, IV) CRC, we only successfully extracted accuracy data from 3 studies, including 1 study which lacked true positive cases and therefore was excluded by statistical software. Due to the limited number of studies (2 studies only), we did not continue the pooling of accuracy data in early-stage CRC group and the comparison between early- and late-stage CRC.

Subgroup analysis was also conducted according to techniques used for cfDNA samples. ARMS was excluded from the subgroup analysis because only 1 study used this technique [20]. After pooling, digital PCR showed higher sensitivity [0.81 (95%CI: 0.78–0.85)] but slightly lower specificity [0.85 (95%CI: 0.83–0.88)], compared to NGS [0.65 (95%CI: 0.59–0.71), 0.88 (95%CI: 0.85–0.91), respectively] (see Table 3). Pooled DOR of digital PCR [29.18 (95%CI: 11.79–72.25)] and AUC of SROC curve (0.9067) were also higher than that of NGS [14.61 (95%CI: 9.78–21.84) and 0.8574]. In subtypes of NGS, a commercial Guardant360 (Guardant Health) NGS panel showed slightly higher sensitivity, specificity, and DOR than the overall

accuracy of NGS (Table 3). In subtypes of digital PCR, BEAMing showed higher sensitivity, specificity, and DOR than digital droplet PCR (see Table 3). Other subtypes were excluded from the analysis due to limited number of studies.

In addition, we also conducted subgroup analysis across different regions of the studies. Results showed the highest sensitivity [0.80 (95%CI: 0.77–0.83)] and DOR [32.97 (95%CI: 13.63–79.79)] in studies from Europe, and the lowest sensitivity in studies from America [0.66 (95%CI: 0.56–0.75)] and lowest DOR [13.87 (95%CI: 7–27.48)] in studies from Asia (see Table 3). The specificity was highest in America [0.92 (95%CI: 0.86–0.96)] and lowest in Asia [0.81 (95%CI: 0.77–0.85)]. After looking into the techniques used for cfDNA samples across different regions, we found that majority of the studies from Europe (14 out of 18) used digital PCR, while majority of the studies from Asia (8 of the 11) and all studies from America (4 out of 4) used NGS instead. Therefore, we further grouped the studies by both region and techniques used for cfDNA samples. Interestingly, after taking techniques into consideration, the sensitivity of NGS was similar across different regions, and sensitivity and DOR of digital PCR were also similar between Europe and Asia (Table 3). Those results indicate that the difference in accuracy among different regions of the studies may be partially explained by the significant difference in the techniques used for cfDNA samples across the regions, although we did observe differences in specificity, and PLR (and also DOR in NGS) across the regions. Further investigation on the subtypes of digital PCR revealed that in studies from Europe, BEAMing had higher sensitivity, specificity, and DOR, compared to digital droplet PCR (see Table 3). Due to limited number of studies, we did not further compare the accuracy of BEAMing and digital droplet PCR in Asia and America, or different panels of next-generation sequencing in all the three regions.

Since this study is investigating diagnostic accuracy, we used Deek's funnel plot asymmetry test to evaluate publication bias (see Deek's funnel plot in Fig 3). The test results indicated no significant publication bias (*P* = 0.12).

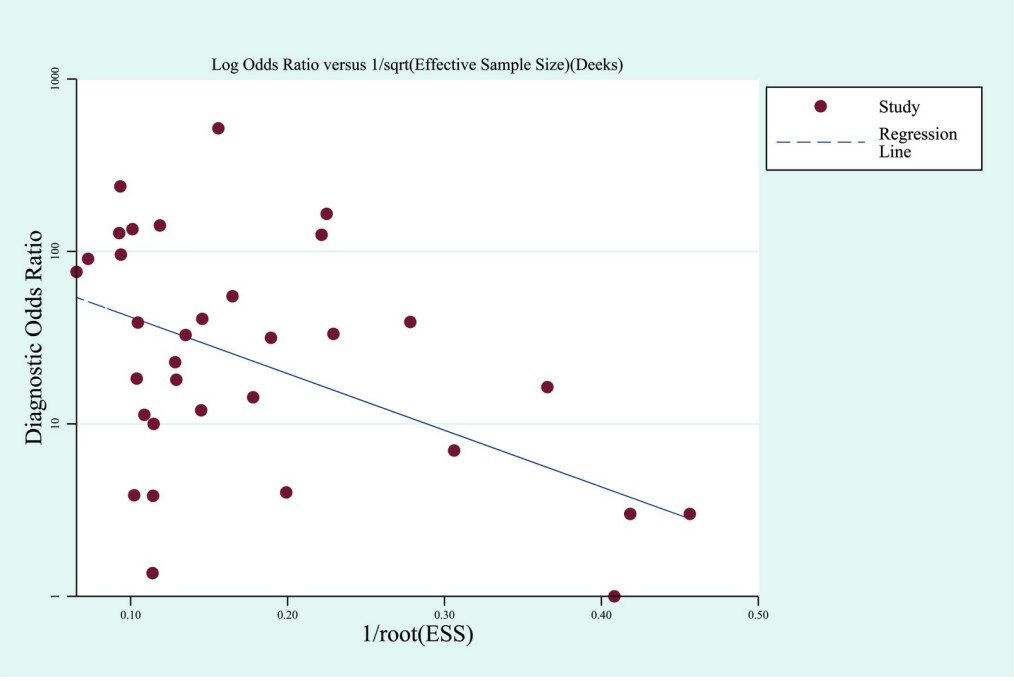

**Fig 3. Deek's funnel plot.**

## Discussion

Since anti-EGFR therapies (cetuximab and panitumumab) showed benefit only in *RAS* wild-type mCRC patients, the precise measurement of *RAS* mutation status in tumor is very important for the success of the targeted therapy [9,10]. Tumor tissue is commonly used for the detection of *RAS* mutation, but in some mCRC patients, tumor tissue is not available. Liquid biopsy sample (or cfDNA) has emerged as an alternative for the determination of *RAS* mutation status [15,16]. However, its accuracy needs to be validated using paired tissue sample as reference ("gold standard").

Many investigations have validated the accuracy of *KRAS*/All-*RAS* mutation detection using liquid biopsy samples. Traditional techniques (e.g. PCR, direct sequencing) were mostly used in the early investigations [56]. In recent years, most of the studies used more sensitive methods, including digital PCR, NGS, and ARMS techniques, and this systemic review and meta-analysis focused on those studies. Thirty-three eligible studies have been involved in our study after database searching and screening. After pooling, the overall sensitivity and specificity of *KRAS* mutation detection using cfDNA samples were 77% and 87%, respectively. The important indicator of diagnostic test [57], DOR, was 23.96, and AUC of SROC curve was 0.8992. Those results suggest an overall high diagnostic accuracy of the *KRA*S mutation detection using cfDNA samples. Previous meta-analysis by Xie et al investigated diagnostic accuracy of *KRAS* mutation detection using ctDNA and the pooled sensitivity, specificity, and DOR were 63.7%, 94.3%, and 37.883, respectively [56]. Our meta-analysis revealed higher sensitivity but lower specificity possibly due to the different diagnostic techniques investigated. Majority of the studies in the study by Xie et al used ARMS or PCR for the detection of *KRAS* mutation in ctDNA, while our study focused more on digital PCR and NGS which were shown to have higher sensitivity than conventional PCR [58,59].

Since significant inter-study heterogeneity was found during the pooling, we further studied its possible sources. We did not observe significant threshold effect, and meta-regression analysis also indicated no association between inter-study heterogeneity and the 5 covariates in our study (technique used for cfDNA samples, technique used for tissue samples, region, race, and type of CRC). We then performed subgroup analysis. After separating and pooling of the results between primary and metastatic CRC patients, *KRAS* mutation detection using cfDNA samples showed higher sensitivity (79%), specificity (88%), DOR (29.17), and AUC of SROC curve (0.9045) in mCRC cases, compared to primary CRC patients (57%, 73%, 10.81, and 0.7304, respectively), indicating that this method might be more suitable for mCRC patients. Among the three testing platforms involved in this meta-analysis, ARMS was excluded from the subgroup analysis because of the limited number of study. Comparison between NGS and digital PCR showed higher sensitivity (81%), DOR (29.18), and AUC of SROC curve (0.9067) in digital PCR compared to NGS (65%, 14.61, and 0.8574), which indicates higher accuracy of digital PCR. Further analysis on the subtypes of the techniques showed that Guardant360 NGS panel had only slightly higher sensitivity, specificity, and DOR compared to overall accuracy of NGS (Table 3). In subtypes of digital PCR, BEAMing showed higher sensitivity (87%), specificity (90%), and DOR (50.96) compared to digital droplet PCR. Even after we limited the region of the studies in Europe, BEAMing also showed better accuracy than digital droplet PCR. Those results indicate that BEAMing is a preferable technique for *KRAS* mutation detection using liquid biopsy samples of CRC patients. Comparison between different regions of the studies showed highest sensitivity (80%) and DOR (32.97) in Europe, compared to Asia and America. Further analysis showed that the accuracy was similar across the three regions when the same type of technique was used, indicating that those differences in accuracy among different regions could be partially due to the different techniques

used. Digital PCR showed higher accuracy compared to NGS (Table 3), which may have led to the overall higher accuracy in Europe where digital PCR is more used in the studies. In addition, the difference in diagnostic accuracy between subgroups might partially explain the heterogeneity among the studies. Publication bias was also investigated using Deek's funnel plot asymmetry test, and results showed no significant publication bias.

Other than the overall or subgroup analyses on the accuracy of *KRAS* mutation detection using liquid biopsy samples as mentioned above, we did observe a wide range of disparities in accuracy among the studies involved in this systemic review. Even in studies using the same methods for plasma (Guardant360) and tissue (FoundationOne) samples, the sensitivity and specificity varied greatly (59.5% – 75.9% and 87.2% – 97.8%, respectively) [19,35,36]. Similar disparities were also observed in studies using commercial BEAMing kit (OncoBEAM™ RAS CRC Kit), although different types of methods were used for tissue samples in those studies. Considering that the commercial NGS panels (Guardant360 and FoundationOne) and Onco-BEAM™ RAS CRC Kit should be well standardized and optimized and all the patient populations in the studies were from the same region (America for Guardant360, or Europe for OncoBEAM™ RAS CRC Kit), the possible sources of those disparities might be from small size of the patient cohort (61–76 patients/study for Guardant360 and 6–236 patients/study for OncoBEAM™ RAS CRC Kit), or differences in how the experiment and data analysis were performed. Although the exact sources are still unknown, those disparities indicate an urgent need in further standardization and optimization of those techniques. On the other hand, the concordance rate of the studies was also not satisfactory. The concordance rate ranged from 60% [29,53] to 97.5% [14], and nearly 40% (13/33) of the studies showed a concordance rate lower than 80%. Those results indicate risks of misdiagnosis using liquid biopsy to detect *KRAS* mutation in CRC patients, and *KRAS* testing results from liquid biopsy samples have to be handled carefully and only be used when tissue samples are not available. Standardization and further optimization of the techniques are needed to hopefully increase the accuracy of the *KRAS* mutation testing using liquid biopsy samples.

Due to its better availability, quick results turnarounds, and minimal-invasiveness, liquid biopsy has been extensively studied for its use in early detection of cancer, prediction of patient prognosis, and monitoring of disease [15]. Several of the studies involved in this systemic review also performed serial monitoring of ctDNA in colorectal cancer patients. Kim et al [32] collected serial plasma samples of two CRC cancer patients during treatment of cetuximab and observed newly-emerged *KRAS* mutations in ctDNA results 1.5 months before radiologic progression. Choi et al [36] monitored ctDNA in serial blood samples from CRC patients on anti-EGFR therapy using Guardant360 NGS panel, and observed multiple emerging genetic alterations associated with treatment resistance. Sun et al [39] used a customized 85-gene NGS panel to monitor ctDNA of early-stage CRC patients for 6 months following surgery, and observed decrease in driver mutation in half of the patients after surgery and increase of *TP53* and *PIK3CA* mutations in a patient with liver metastasis. Vidal et al [45] used OncoBEAM™ RAS CRC Kit to monitor *RAS* mutations in blood samples from mCRC patients during their anti-EGFR treatment, and found that *RAS* mutations in ctDNA mirrored the response to treatment. In addition, several on-going prospective clinical trials are also investigating the use of cfDNA in predicting treatment or relapse in earlier stage (I, II or III) or resectable CRC (e.g. NCT04068103, NCT04486378, NCT04264702, NCT04050345). Although many encouraging results were shown in those studies, solid evidence from prospective randomized clinical trials is required before the complete adoption of liquid biopsy in clinical practice [15,60]. From the results of this systemic review and meta-analysis, we also observed wide disparities among the accuracy of *KRAS* mutation detection using liquid biopsy, even in well standardized and optimized commercial kits (as described above). In addition, pooled accuracy for *KRAS* mutation

detection using liquid biopsy is suboptimal in primary CRC patients (Table 3), indicating significant risk of misdiagnosis. Oncologists should still be very cautious when using liquid biopsy results to guide clinical practices, before evidence from clinical trials proves excellent accuracy of *KRAS* mutation detection using liquid biopsy, or clear benefit in patient survival in ctDNA/liquid biopsy-guided targeted therapies.

## Conclusions

In all, our study showed that NGS, digital PCR, and ARMS techniques had overall high accuracy in detecting *KRAS* mutation in liquid biopsy samples. The results could be used to guide anti-EGFR therapy in CRC patients with no available tumor tissue samples, but need to be handled carefully considering the potential risk of discordance and misdiagnosis. *KRAS* mutation detection in liquid biopsy samples had higher accuracy in mCRC patients compared to primary CRC patients, and is therefore more recommended in mCRC patients. Due to its better availability, liquid biopsy could be helpful in early detection and monitoring of CRC, and prediction of patient prognosis, and many studies and clinical trials are investigating its possible roles in those applications. However, oncologists should still be very cautious when using liquid biopsy result in guiding clinical practices, before solid evidence from prospective randomized clinical trials proves its usefulness. Digital PCR also showed higher accuracy than NGS, and among their subtypes, BEAMing showed the highest accuracy, and is recommended for *KRAS* mutation detection in liquid biopsy samples. Limitation of the study may include that number of studies involved in some subgroups (e.g. primary CRC group) is still quite small, and the results should be handled carefully. In addition, although the accuracy of difference techniques does not differ much when analyzing the highly abundant tumor-derived DNA in tissue samples, different techniques used in the reference group (tumor tissue samples) may still cause potential bias. Other potential variations between the studies (e.g. different patient cohorts, different supplier of experimental reagents, and etc.) may also cause bias to the results. Large prospective randomized clinical trials are needed to further convince the accuracy and usefulness of *KRAS* mutation detection using cfDNA/liquid biopsy samples in clinical practices.

## Supporting information

**S1 Table. Search strategy.**
(DOCX)

**S1 File. PRISMA checklist.**
(DOC)

**S2 File. Data extracted from eligible studies.** Data were also deposited in Systematic Review Data Repository (SRDR): https://srdr.ahrq.gov/projects/1639.
(XLSX)

## Author Contributions

**Conceptualization:** Peng Ye, Yuanyuan Wei.

**Data curation:** Peng Ye, Peiling Cai.

**Funding acquisition:** Yuanyuan Wei.

**Methodology:** Peng Ye, Jing Xie.

**Resources:** Peng Ye, Peiling Cai.

**Supervision:** Yuanyuan Wei.

**Writing – original draft:** Peng Ye.

**Writing – review & editing:** Peiling Cai, Jing Xie, Yuanyuan Wei.

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
