## [Decision Letter · Decision Letter 0]

8 Jan 2021

PONE-D-20-31082

The diagnostic accuracy of digital PCR, ARMS and NGS for detecting KRAS mutation in cell-free DNA of patients with colorectal cancer: a systematic review and meta-analysis

PLOS ONE

Dear Dr. Ye,

Thank you for submitting your manuscript to PLOS ONE. After careful consideration, we feel that it has merit but does not fully meet PLOS ONE’s publication criteria as it currently stands. Therefore, we invite you to submit a revised version of the manuscript that addresses the points raised during the review process.

As noted by the reviewer, the authors should better describe the assay systems used and the differences in technical approaches.  They should also include the full names, not just the abbreviations such as for ARMS (Amplification Refractory Mutation System) and what assay versions were used, such as the new SuperARMS.  While some of the studies they discussed may have used KRAS alone, others included NRAS and this should be noted in the review. Finally, as suggested the article is written in English, but this is not the authors’ first language and a through review of the grammar is needed.

We look forward to receiving your revised manuscript.

Kind regards,

Anthony F. Shields, M.D., Ph.D.

Academic Editor

PLOS ONE

Journal Requirements:

2.) Thank you for including the statement that 'Literature research was performed independently by PY and PC in June 2020'. Please revise this statement to clarify whether all databases were searched from inception, or if there were any limits placed on the publication dates in your search.

3.) At this time, we ask that you please provide the full search strategy and search terms for at least one database used as Supplementary Information.

4.) In the Methods section, please provide the methodology used for data extraction, including the specific reporting items that were extracted from the included studies.

Reviewers' comments:

Reviewer's Responses to Questions

**Comments to the Author**

1. Is the manuscript technically sound, and do the data support the conclusions?

Reviewer #1: Partly

Reviewer #2: Yes

2. Has the statistical analysis been performed appropriately and rigorously? 

Reviewer #1: I Don't Know

Reviewer #2: Yes

3. Have the authors made all data underlying the findings in their manuscript fully available?

Reviewer #1: Yes

Reviewer #2: Yes

4. Is the manuscript presented in an intelligible fashion and written in standard English?

Reviewer #1: Yes

Reviewer #2: No

5. Review Comments to the Author

Reviewer #1: The investigators performed literature review to identify studies that have examined KRAS status using cell free DNA. They performed meta-analysis of these data and present the results here.

The cited references appear to be during or after 2016, shortly after a time when expansion of standard tissue-based platforms adapted to the concept of All-RAS sequencing (more isoforms and testing of NRAS as well). Please clarify confirmation that the papers using meta-analysis adhered to expanded testing, and also clarify whether NRAS (while much less prevalent, 5-10% in CRC) was also assessed in any or all the studies. Also, it would be helpful to clarify whether there were notable differences in testing approach in relation to accuracy by region/country of testing, and whether any potential differences could be accounted for by significant differences in testing approaches.

One of the proposed advantages of cell free DNA is more accurate identification of tumor cell heterogeneity, including potential mixed populations of KRAS mutant and KRAS wild type cells. Not all the studies divided examined here, reported what percentage of samples yielded positive results for both within the same specimens.

The authors pool results across many different studies, using many different approaches, and the final pulled result does not mean much considering the wide range. For presentation of statistical analysis, it would be preferable to emphasize more the wide range and median values, with a subset analyses to report common themes and approaches that appear to be reproducible and precise (and also hopefully providing accuracy ). There are many disparate results reported by a number of these papers, including a wide range of sensitivity of assays, even when using the same method (eg NGS), and differences between cell free DNA-based assessment versus tissue-based assessment. The latter point is well documented, and more in-depth comments about the wide range of disparities should be included in the discussion section. Concordance rates are poor, yet many oncologists worldwide depend heavily on liquid-based biopsy at assessment for determination of using EGFR inhibitor treatments. What is the authors’ perspective on how this practice should change, and how, based on the results of this meta-analysis? I encourage the investigators to leverage this manuscript as a platform to propose change in the field for better and more accurate use of these tests.

The use of serial assessment of KRAS and other markers over time in colorectal cancer has been promoted without strong evidence that this changes management on a large scale, nor results in improvement of overall survival. Can the authors comment on perspective on this angle, noting also that there are multiple prospective trials examining the use of cell free DNA in general in the postoperative setting for earlier stage/resectable colorectal cancers (in the U.S.).

On page 18, it is mentioned that some studies assessing KRAS were performed in patients with primary colorectal cancer; please clarify why this was done in the studies, as this type of assessment is not routinely indicated in nonmetastatic patients. As the data are there, it would also be helpful to discuss any notable differences in sensitivity, and/or concordance, between primary and metastatic tumors that were assessed in relation to cell free DNA, and early-stage versus later stage patients.

Reviewer #2: In this review/meta-analysis, Peng Ye tried to compare different methods used in detecting KRAS mutations in plasma of CRC patients. Although the authors put a lot of efforts into, there remain some issues/changes to be done:

1.) Improve English! There are several grammer errors!

2.) In this study, the authors are comparing PCR, NGS and ARMS. However, they are describing these methods only superficial. Thus, in the introduction section the authors should describe these techniques (what is better in one to another method? costs issues? ecc...).

3.) The authors are describing that only KRAS is important in the selection for anti-EGFR treatment. This is not correct! The authors should focus, or at least discuss, that next to KRAS mutations also NRAS mutations are very important to select the best treatment for our CRC patients.

4.) The authors are concluding that liquid biopsy may be only important in the metastatic stage, however, also in limited stages liquid biopys may be helpful in monitoring disease or detecting early relapse. Please, discuss this fact.

6. PLOS authors have the option to publish the peer review history of their article (what does this mean?). If published, this will include your full peer review and any attached files.

Reviewer #1: No

Reviewer #2: No

---

## [Author Response · Author response to Decision Letter 0]

1 Feb 2021

Comments from Editor:

1) As noted by the reviewer, the authors should better describe the assay systems used and the differences in technical approaches. 

Response: Many thanks for the suggestions. We have revised the manuscript and included more descriptions on the three assay systems and their advantages and disadvantages. Please find the revision in the 3rd paragraph of the Introduction section (page 5 of the Revised Manuscript with Track Changes).

2) They should also include the full names, not just the abbreviations such as for ARMS (Amplification Refractory Mutation System) and what assay versions were used, such as the new SuperARMS. 

Response: Many thanks for the suggestions. We have revised the manuscript and included the detailed type/version of techniques used in the studies. Please find the revision in Table 1 on page 10-17 in the Revised Manuscript with Track Changes. In addition, we also revised the names of techniques and avoided usage of abbreviations. Please find the revisions in the Revised Manuscript with Track Changes.

3) While some of the studies they discussed may have used KRAS alone, others included NRAS and this should be noted in the review. 

Response: Thanks very much for the suggestion. In the 33 eligible studies, there were in all 20 studies which targeted All-RAS, or expanded isoforms of KRAS. Most of those studies reported KRAS sequencing results separately from NRAS/HRAS, while 6 studies reported only All-RAS results. In this case, the NRAS testing results were included in the systematic review and meta-analysis. We have revised the manuscript accordingly. Please find the revisions in the 1st paragraph of “Review of eligible publications” section (page 9 of the Revised Manuscript with Track Changes).

4) Finally, as suggested the article is written in English, but this is not the authors’ first language and a through review of the grammar is needed.

Response: Very sorry for the grammar errors. We have carefully proofread the manuscript and tried our best to remove the grammar errors. Please find the revisions in the Revised Manuscript with Track Changes.

Additional requirements:

 Response: Many thanks for your comments. We have carefully checked and ensured that our manuscript meets PLOS ONE’s style requirement.

2.) Thank you for including the statement that 'Literature research was performed independently by PY and PC in June 2020'. Please revise this statement to clarify whether all databases were searched from inception, or if there were any limits placed on the publication dates in your search.

Response: Thanks very much for your suggestion. We did not place any limitation on the publication dates in the literature search. Please find the revised statement in the 1st paragraph in “Literature searching and selection of publication” section (page 6 of the Revised Manuscript with Track Changes).

3.) At this time, we ask that you please provide the full search strategy and search terms for at least one database used as Supplementary Information.

Response: Thanks very much for your suggestion. We have included the detailed search strategy as S1 Table. Please find it in the supporting information.

4.) In the Methods section, please provide the methodology used for data extraction, including the specific reporting items that were extracted from the included studies.

Response: Thanks very much for your comments. We have revised the manuscript and included all the reporting items used in our data extraction. Please find the revisions in the 2nd paragraph of “Literature searching and selection of publication” section (page 6-8 of the Revised Manuscript with Track Changes).

 

Comments from Review #1:

The investigators performed literature review to identify studies that have examined KRAS status using cell free DNA. They performed meta-analysis of these data and present the results here.

1) The cited references appear to be during or after 2016, shortly after a time when expansion of standard tissue-based platforms adapted to the concept of All-RAS sequencing (more isoforms and testing of NRAS as well). Please clarify confirmation that the papers using meta-analysis adhered to expanded testing, and also clarify whether NRAS (while much less prevalent, 5-10% in CRC) was also assessed in any or all the studies. 

Response: Thanks very much for your suggestions. In our review, 20 eligible studies performed All-RAS sequencing with expanded isoforms of the genes. Most of the studies reported KRAS testing results separately, or ignored NRAS/HRAS testing results. The rest 6 studies reported All-RAS sequencing results only, and in this case, the NRAS testing results were included in the subsequent systematic review and meta-analysis. We also revised the manuscript to describe those details. Please find them in the 1st paragraph of “Review of eligible publications” section (page 9 of Revised Manuscript with Track Changes).

2) Also, it would be helpful to clarify whether there were notable differences in testing approach in relation to accuracy by region/country of testing, and whether any potential differences could be accounted for by significant differences in testing approaches.

Response: Thanks very much for the helpful suggestions. We performed meta-analysis across different regions in studies using next-generation sequencing or digital polymerase chain reaction, and the sensitivity was similar across different regions for next-generation sequencing, and sensitivity and diagnostic odds ratio were similar between Europe and Asia, indicating that the differences in accuracy among different regions could be partially explained by significant difference in techniques used in those regions. We also inserted those results and discussions in the 5th paragraph of “Meta-analysis of the accuracy of KRAS mutation detection using cfDNA samples” section (page 29-30 of the Revised Manuscript with Track Changes).

3) One of the proposed advantages of cell free DNA is more accurate identification of tumor cell heterogeneity, including potential mixed populations of KRAS mutant and KRAS wild type cells. Not all the studies divided examined here, reported what percentage of samples yielded positive results for both within the same specimens.

Response: Thanks very much for the comments. We agree that the tumor heterogeneity is not a major focus of this meta-analysis or the references of this manuscript. The expression was removed accordingly. Please find the revision in the 2nd paragraph of Introduction section (page 5 of the Revised Manuscript with Track Changes).

4) The authors pool results across many different studies, using many different approaches, and the final pulled result does not mean much considering the wide range. For presentation of statistical analysis, it would be preferable to emphasize more the wide range and median values, with a subset analyses to report common themes and approaches that appear to be reproducible and precise (and also hopefully providing accuracy ). 

Response: Thanks very much for your suggestions. We have put more effort on the subgroup analysis. In addition to the comparison among different regions in studies using next-generation sequencing or digital polymerase chain reaction, we also investigated the accuracy of subtypes of next-generation sequencing (Guardant360 panel) and digital polymerase chain reaction (Beads, Emulsion, Amplification and Magnetics, and digital droplet polymerase chain reaction). In addition, we emphasized more on the heterogeneity among the studies and also addressed that we focus more on the possible sources of heterogeneity and subgroup analysis. Please find the revisions in the 2nd and 4th paragraph of “Meta-analysis of the accuracy of KRAS mutation detection using cfDNA samples” section (page 25, 29 of the Revised Manuscript with Track Changes), and Table 3 (page 26-28 of the Revised Manuscript with Track Changes).

5) There are many disparate results reported by a number of these papers, including a wide range of sensitivity of assays, even when using the same method (eg NGS), and differences between cell free DNA-based assessment versus tissue-based assessment. The latter point is well documented, and more in-depth comments about the wide range of disparities should be included in the discussion section. 

Response: Thanks very much for your helpful suggestions. We have looked into the wide range of disparities among the studies, and inserted comments in the discussion section. Please find the revision in the 4th paragraph of Discussion section (page 34 of the Revised Manuscript with Track Changes).

6) Concordance rates are poor, yet many oncologists worldwide depend heavily on liquid-based biopsy at assessment for determination of using EGFR inhibitor treatments. What is the authors’ perspective on how this practice should change, and how, based on the results of this meta-analysis? I encourage the investigators to leverage this manuscript as a platform to propose change in the field for better and more accurate use of these tests.

Response: Thanks very much for your helpful suggestions. Accordingly, we also commented on the poor concordance rate and proposed possible ways to limit the risk of misdiagnosis. Please find the comments in the 4th paragraph of Discussion section, after the comments for wide range of disparities among the studies (page 34-35 of the Revised Manuscript with Track Changes).

7) The use of serial assessment of KRAS and other markers over time in colorectal cancer has been promoted without strong evidence that this changes management on a large scale, nor results in improvement of overall survival. Can the authors comment on perspective on this angle, noting also that there are multiple prospective trials examining the use of cell free DNA in general in the postoperative setting for earlier stage/resectable colorectal cancers (in the U.S.).

Response: Thanks very much for your helpful suggestions. We have looked into the serial assessment of KRAS using liquid biopsy samples. Please find our comments in the 5th paragraph of Discussion section in the revised manuscript (page 35-36 of the Revised Manuscript with Track Changes).

8) On page 18, it is mentioned that some studies assessing KRAS were performed in patients with primary colorectal cancer; please clarify why this was done in the studies, as this type of assessment is not routinely indicated in nonmetastatic patients. 

Response: Thanks very much for your suggestions. We have looked into the reasons for the involvement of nonmetastatic patients in those studies, and inserted them in the start of the 3rd paragraph in “Meta-analysis of the accuracy of KRAS mutation detection using cfDNA samples” in Results section (page 26 of the Revised Manuscript with Track Changes).

9) As the data are there, it would also be helpful to discuss any notable differences in sensitivity, and/or concordance, between primary and metastatic tumors that were assessed in relation to cell free DNA, and early-stage versus later stage patients.

Response: Thanks very much for your suggestions. We compared the accuracy data between primary and metastatic tumors in relation to cell free DNA, and the results could be found in the 3rd paragraph in “Meta-analysis of the accuracy of KRAS mutation detection using cfDNA samples” in Results section (page 26 of the Revised Manuscript with Track Changes), and the 3rd paragraph (page 32 of the Revised Manuscript with Track Changes) and 5th paragraph of Discussion section (line 3-5 on page 36 of the Revised Manuscript with Track Changes). In addition, following your advices, we tried to extract the accuracy data from early-stage CRC patients. But unfortunately, data could only be extracted from 3 studies, including 1 study which lacked true positive cases and was therefore excluded by the statistical software. The number of the rest studies (2 studies) was too small for meta-analysis. Therefore, we did not continue the comparison between early-stage and late-stage patients. Those results were also described at the end of 3rd paragraph in “Meta-analysis of the accuracy of KRAS mutation detection using cfDNA samples” in Results section (page 26 of the Revised Manuscript with Track Changes).

 

Comments from Review #2:

In this review/meta-analysis, Peng Ye tried to compare different methods used in detecting KRAS mutations in plasma of CRC patients. Although the authors put a lot of efforts into, there remain some issues/changes to be done:

1.) Improve English! There are several grammer errors!

Response: Very sorry for the grammar errors. We have carefully proofread the manuscript and tried our best to remove the grammar errors. Please find the revisions in the Revised Manuscript with Track Changes.

2.) In this study, the authors are comparing PCR, NGS and ARMS. However, they are describing these methods only superficial. Thus, in the introduction section the authors should describe these techniques (what is better in one to another method? costs issues? ecc...).

Response: Thanks very much for the suggestions. We have revised the introduction section and described more advantages and disadvantages of those techniques. Please find the revision in the 3rd paragraph of Introduction section (page 5-6 of the Revised Manuscript with Track Changes).

3.) The authors are describing that only KRAS is important in the selection for anti-EGFR treatment. This is not correct! The authors should focus, or at least discuss, that next to KRAS mutations also NRAS mutations are very important to select the best treatment for our CRC patients.

Response: Thanks very much for your comments. We have revised the introduction section and emphasized the importance of NRAS in the anti-EGFR treatment for CRC patients. Please find the revision in the 1st paragraph of Introduction section (page 4 of the Revised Manuscript with Track Changes).

4.) The authors are concluding that liquid biopsy may be only important in the metastatic stage, however, also in limited stages liquid biopys may be helpful in monitoring disease or detecting early relapse. Please, discuss this fact.

Response: Many thanks for your comments. We discussed more about the use of liquid biopsy in early detection and monitoring of the disease and prediction of patients prognosis. Please find the revision in the 5th paragraph of Discussion section (page 35-36 of the Revised Manuscript with Track Changes), and in Conclusion section (at end of page 36 of the Revised Manuscript with Track Changes).

---

## [Editor Report · Decision Letter 1]

18 Feb 2021

PONE-D-20-31082R1

The diagnostic accuracy of digital PCR, ARMS and NGS for detecting KRAS mutation in cell-free DNA of patients with colorectal cancer: a systematic review and meta-analysis

PLOS ONE

Dear Dr. Ye,

Thank you for submitting your manuscript to PLOS ONE. After careful consideration, we feel that it has merit but does not fully meet PLOS ONE’s publication criteria as it currently stands. Therefore, we invite you to submit a revised version of the manuscript that addresses the points raised during the review process.

The authors misinterpreted the editor's suggestion to "include the full names, not just the abbreviations such as for ARMS" etc. Terms such as ARMS, PCR, and NGS need to be defined at the start, but abbreviations can be used after that. Please correct that. Otherwise, the manuscript is much improved and acceptable for publication.

A marked-up copy of your manuscript that highlights changes made to the original version. You should upload this as a separate file labeled 'Revised Manuscript with Track Changes'.An unmarked version of your revised paper without tracked changes. You should upload this as a separate file labeled 'Manuscript'.

We look forward to receiving your revised manuscript.

Kind regards,

Anthony F. Shields, M.D., Ph.D.

Academic Editor

PLOS ONE

Additional Editor Comments (if provided):

The authors misinterpreted the editor's suggestion to "include the full names, not just the abbreviations such as for ARMS" etc. Terms such as ARMS, PCR, and NGS need to be defined at the start, but abbreviations can be used after that. Please correct that. Otherwise, the manuscript is much improved and acceptable for publication.

---

## [Author Response · Author response to Decision Letter 1]

18 Feb 2021

Dear Editor,

Thank you very much for your decision letter and advice on our manuscript entitled “The diagnostic accuracy of digital PCR, ARMS and NGS for detecting KRAS mutation in cell-free DNA of patients with colorectal cancer: a systematic review and meta-analysis” (PONE-D-20-31082R1) to be considered for publication in PLOS ONE. 

Please see our response to your comments below.

1. The authors misinterpreted the editor's suggestion to "include the full names, not just the abbreviations such as for ARMS" etc. Terms such as ARMS, PCR, and NGS need to be defined at the start, but abbreviations can be used after that. Please correct that. Otherwise, the manuscript is much improved and acceptable for publication.

Response: Very sorry for the misinterpretation and troubles made. Following your suggestions, we have changed the full names of ARMS, PCR, NGS, and BEAMing into abbreviations after defining them at first use. Please find the corrections in the Revised Manuscript with Track Changes.

Look forward to hearing from you soon.

Sincerely yours,

Peng Ye

Chengdu University, Chengdu 610106, P.R.China

Tel: (86)18602885572

E-mail: yepeng@cdu.edu.cn

---

## [Editor Report · Decision Letter 2]

5 Mar 2021

The diagnostic accuracy of digital PCR, ARMS and NGS for detecting KRAS mutation in cell-free DNA of patients with colorectal cancer: a systematic review and meta-analysis

PONE-D-20-31082R2

Dear Dr. Ye,

We’re pleased to inform you that your manuscript has been judged scientifically suitable for publication and will be formally accepted for publication once it meets all outstanding technical requirements.

Kind regards,

Anthony F. Shields, M.D., Ph.D.

Academic Editor

PLOS ONE

---

## [Editor Report · Acceptance letter]

17 Mar 2021

PONE-D-20-31082R2 

The diagnostic accuracy of digital PCR, ARMS and NGS for detecting KRAS mutation in cell-free DNA of patients with colorectal cancer: a systematic review and meta-analysis 

Dear Dr. Ye:

I'm pleased to inform you that your manuscript has been deemed suitable for publication in PLOS ONE. Congratulations! Your manuscript is now with our production department. 

Kind regards, 

on behalf of

Dr. Anthony F. Shields 

Academic Editor

PLOS ONE